# Unorthodox PCNA Binding by Chromatin Assembly Factor 1

**DOI:** 10.3390/ijms231911099

**Published:** 2022-09-21

**Authors:** Amogh Gopinathan Nair, Nick Rabas, Sara Lejon, Caleb Homiski, Michael J. Osborne, Normand Cyr, Aleksandr Sverzhinsky, Thomas Melendy, John M. Pascal, Ernest D. Laue, Katherine L. B. Borden, James G. Omichinski, Alain Verreault

**Affiliations:** 1Institute for Research in Immunology and Cancer, University of Montreal, Montreal, QC H3T 1J4, Canada; 2Molecular Biology Program, University of Montreal, Montreal, QC H3T 1J4, Canada; 3Department of Biochemistry, University of Cambridge, 80 Tennis Court Road, Cambridge CB2 1GA, UK; 4Departments of Biochemistry and Microbiology & Immunology, University at Buffalo Jacobs School of Medicine & Biomedical Sciences, 955 Main Street, Buffalo, NY 14210, USA; 5Department of Biochemistry and Molecular Medicine, University of Montreal, Montreal, QC H3C 3J7, Canada; 6Department of Pathology and Cell Biology, University of Montreal, Montreal, QC H3T 1J4, Canada

**Keywords:** CHAF1A, PCNA, PIP-Box, DNA replication, CAF-1 p150, chromatin assembly, SEC-SAXS, NMR

## Abstract

The eukaryotic DNA replication fork is a hub of enzymes that continuously act to synthesize DNA, propagate DNA methylation and other epigenetic marks, perform quality control, repair nascent DNA, and package this DNA into chromatin. Many of the enzymes involved in these spatiotemporally correlated processes perform their functions by binding to proliferating cell nuclear antigen (PCNA). A long-standing question has been how the plethora of PCNA-binding enzymes exert their activities without interfering with each other. As a first step towards deciphering this complex regulation, we studied how Chromatin Assembly Factor 1 (CAF-1) binds to PCNA. We demonstrate that CAF-1 binds to PCNA in a heretofore uncharacterized manner that depends upon a cation-pi (π) interaction. An arginine residue, conserved among CAF-1 homologs but absent from other PCNA-binding proteins, inserts into the hydrophobic pocket normally occupied by proteins that contain canonical PCNA interaction peptides (PIPs). Mutation of this arginine disrupts the ability of CAF-1 to bind PCNA and to assemble chromatin. The PIP of the CAF-1 p150 subunit resides at the extreme C-terminus of an apparent long α-helix (119 amino acids) that has been reported to bind DNA. The length of that helix and the presence of a PIP at the C-terminus are evolutionarily conserved among numerous species, ranging from yeast to humans. This arrangement of a very long DNA-binding coiled-coil that terminates in PIPs may serve to coordinate DNA and PCNA binding by CAF-1.

## 1. Introduction

Ahead of the replication fork, pre-existing nucleosomes are dismantled and temporarily dissociated into (H3-H4)_2_ tetramers and H2A-H2B dimers [1,2]. The tetramers are segregated to the two nascent sister chromatids in a quasi-stochastic manner [1,2]. Nucleosome-free gaps generated by DNA duplication are almost immediately filled in by newly synthesized H3-H4 deposited onto DNA by a histone chaperone known as Chromatin Assembly Factor 1 (CAF-1). Despite the existence of several histone chaperones, CAF-1 is thus far unique in its ability to mediate chromatin assembly in a manner that is tightly coupled to DNA synthesis [3,4,5]. This unique property of CAF-1 depends upon its ability to bind proliferating cell nuclear antigen (PCNA), a homo-trimeric ring that encircles DNA and serves as a processivity factor for DNA polymerases [4,6,7,8,9]. In addition, at each DNA replication fork, the same surface of each PCNA monomer can engage in interactions with numerous enzymes, including: DNA replication enzymes, CAF-1, DNA methylases (DNMT1) [10], mismatch repair (Msh3 and Msh6) [11], and many others [12,13,14,15].

CAF-1 is composed of three subunits, conserved from yeast to human. In humans, mice, and Xenopus, the three subunits of CAF-1 are p150 (CHAF1A), p60 (CHAF1B), and RbAp48 (RBBP4) [7,16,17]. p150 contains two regions that bind PCNA. The first is a motif located within the N-terminal domain of p150 that we previously referred to as the PCNA binding domain (PBD) (Figure 1A) [18]. Although PBD binds strongly to PCNA in vitro, it does not bear obvious similarity to the large family of PCNA interaction peptides (PIPs), and it is dispensable for replication-coupled chromatin assembly [18]. In contrast, the PIP located internally within the amino acid sequence of p150 (Figure 1A) is unusual because canonical PIPs are generally located at the N- or C-termini of the PCNA-interacting polypeptides. The PIP of p150 shows weaker affinity for PCNA but is nonetheless essential for chromatin assembly [18]. 

The consensus sequence for canonical PIPs is composed of eight residues, Qxxhxxaa, wherein a conserved glutamine is present at position one, an aliphatic residue at position four (leucine, isoleucine, valine or methionine), and two aromatic residues at positions seven and eight (phenylalanine, tyrosine or tryptophan) (Figure 1B) [14]. Several structures have revealed recurring features of the interaction between canonical PIPs and PCNA [19,20,21,22,23,24]. The carbonyl and amino groups of the glutamine side chain of canonical PIP-containing proteins contribute to binding by making several hydrogen bonds with residues in a small surface cavity of PCNA termed the “Q-pocket” found on the front face of PCNA (i.e., the face of the PCNA ring oriented in the direction of DNA synthesis) [19,20,21,22,23,24,25]. This is illustrated by the binding of PCNA to the canonical PIP of human p21 (Figure 1C). The side chain of Q144 is anchored in the so-called Q pocket of PCNA, whereas M147, F150, and Y151 form the three prongs of a hydrophobic plug that binds to PCNA (Figure 1C). The aliphatic and the aromatic residues of canonical PIPs form a short 3_10_ helix and generally interact with PCNA in a manner analogous to that of a plug fitting into a three-pin electrical socket [19,26]. 

We previously reported that the PIP of CAF-1 p150 possesses non-canonical properties [18]. For instance, the CAF-1 PIP has an intrinsic ability to preferentially inhibit nucleosome assembly over DNA synthesis in a replication coupled nucleosome assembly assay [18]. In order to probe this further, we determined the preliminary model of the structure of CAF-1 p150 PIP bound to PCNA based on a molecular replacement solution that could not be refined further. We report here that the CAF-1 PIP binds to PCNA in an unprecedented manner. Our model revealed that a cation-π interaction occurs between PCNA and a conserved arginine in the CAF-1 p150 PIP that is not found in the PIPs of other PCNA-binding proteins. We show that point mutation of this arginine disrupts PCNA binding and chromatin assembly by CAF-1. The CAF-1 PIP is located at the extreme C-terminal end of the KER domain (a domain rich with charged residues), which we showed has an unusually long α-helix that binds to DNA. The length of the helix, and the location of the PIP at its C-terminus are evolutionarily conserved from yeast to humans. We speculate that the very long DNA-binding coiled-coil ending in PIPs may serve to coordinate DNA and PCNA binding by CAF-1.

## 2. Results

### 2.1. Data Based Model of CAF-1 p150 PIP Bound to PCNA 

In order to better understand the molecular basis of CAF-1 binding to PCNA, we determined the crystal model of a complex between a synthetic peptide encompassing the p150 PIP and PCNA. The crystal model was solved using molecular replacement from a previously published PCNA structure [27] (Table 1 and Materials and Methods). PCNA samples were incubated with a 20-amino acid PIP derived from CAF-1 p150. The PIP used for crystallization, IKAE*KAEITRFF*QKPKTPQA (PIP residues in italics), spanned the entire PIP region. 

The crystal model revealed that there were two PCNA homotrimers per asymmetric unit (Figure 2A). Electron density that could not be accounted for by molecular replacement arose from PIPs bound to each monomer in the asymmetric unit. The initial R_work_ and R_free_ values were considerably reduced by modelling in three residues: RFF (residues 426–428 of p150), of the PIP used for crystallization: IKAEKAEIT***RFF***QKPKTPQA (PIP residues for which electron density was attributed are shown in bold italics). To assess the quality of the model, a F_o_-F_c_ difference density map was generated (Figure 2B). The map was calculated by omitting the peptide and showed that the modelled RFF residues were consistent with the electron density unaccounted for by that of PCNA alone.

Our model was consistent with the following general orientation of the PIPs relative to PCNA (Figure 2A). As expected from previous structures of PIPs bound to PCNA, there was one CAF-1 p150 PIP bound to each PCNA monomer. As is for many other PIPs (canonical and non-canonical), the CAF-1 p150 PIPs were bound between the interdomain connector loop (IDCL) and the underlying β-sheet (Figure 2A). An enlarged view of how the p150 PIP is bound to PCNA is illustrated in Figure 2B. The side chains of Arg426, Phe427, and Phe428 were oriented towards PCNA and occupied a surface of PCNA that was essentially the same as that observed in structures of other PCNA-binding peptides. The side chains of Arg426 and Phe427 reside within a hydrophobic pocket on the surface of PCNA, while the side chain of Phe428 contacts the edge of the hydrophobic pocket and is more exposed to solvent (Figure 2C). 

Given the hydrophilic nature of the Arg426 side chain, its insertion within a hydrophobic pocket was unanticipated. However, an inspection of PCNA residues in close proximity to the side chain of Arg426 of CAF-1 p150 suggested that it participates in a cation-π interaction with the tyrosine ring of Tyr 250 from PCNA (Figure 3A). Figure 3B shows other residues that contribute to the hydrophobic environment in which the arginine side chain is anchored by the cation-π interaction.

### 2.2. Structure-Based Mutagenesis of CAF-1 p150

Only three residues (RFF) of the CAF-1 PIP were identified in the model. This was unexpected given the 20-residue peptide utilized for crystallization and the typical 8-residue length of most canonical PIPs. The RFF sequence binds to the same surface of PCNA as canonical and non-canonical PIPs. However, the orientations of the RFF peptide sidechains are different from those of the PIPs of p21, FEN1, p66 subunit of Pol δ, and the error-prone DNA polymerase Pol κ [non-canonical PIP] (Figure 3C,D). 

Phe427 and Phe428 of the RFF peptide were previously shown to be essential for PCNA binding and the nucleosome assembly activity of intact CAF-1 p150 [18]. The novel feature is Arg426, which is not conserved in the PIPs of other PCNA-binding proteins but seemingly plays an important role in the CAF-1 p150 PIP. In fact, the structures of other PIPs bound to PCNA show that the residues corresponding to the arginine of CAF-1 p150 do not contact PCNA. Furthermore, an arginine or a lysine is conserved in that position among CAF-1 p150 homologs from numerous species whose common ancestors lived at least 736 million years ago (Figure 1B). As a result, we concentrated our efforts on the arginine residue of CAF-1 p150, which is involved in a cation-π interaction with PCNA, which is unprecedented among PCNA-binding proteins. 

#### 2.2.1. CAF-1 PIP Binding to PCNA Requires Arg426 p150

We mutated Arg426 of the mouse p150 (equivalent to Arg447 in human p150) to Asp and Ser, commonly found at residues at this position within canonical PIPs (Figure 1B). In addition, neither Ser nor Asp can participate in the cation-π interaction formed by the Arg. 

The wild-type and R426D peptides were expressed as GST fusions in *E.*
*coli*, purified to homogeneity (see Materials and Methods) and used in isothermal titration calorimetry (ITC) assays with human PCNA purified from *E.*
*coli*. When ITC experiments were performed in 10 mM sodium phosphate, pH 7.0, and 10 mM NaCl, we observed binding between the wild-type CAF-1 PIP peptide and PCNA with a *K_d_* ^app^ equivalent to 24 μM and a stoichiometry of 0.83 PIP per PCNA monomer (Figure 4A and Table 2). 

In contrast, we did not observe any heat exchange when we used the CAF-1 PIP R426D peptide (Figure 4A and Table 2). This suggests that the R426D substitution severely abrogates binding to PCNA. This interpretation is consistent with several other lines of evidence described below. We mutated the PCNA tyrosine involved in the cation-π interaction with CAF-1 p150 Arg426. The resulting mutant, PCNA-Y250I was expressed and purified from *E.*
*coli*. Based on size exclusion chromatography, there was no evidence of aggregation that might suggest inappropriate folding of PCNA-Y250I. The wild-type PIP was then titrated into PCNA-Y250I, and there was no detectable heat exchange observed by ITC (Appendix A). 

Taken together, the results obtained with the PIP R426D and the PCNA-Y250I mutants were consistent with the cation-π interaction observed in the crystal model. 

#### 2.2.2. Arginine 426 Is Important for Targeting CAF-1 to DNA Replication Foci Containing PCNA

We next evaluated whether the wild-type CAF-1 p150 and the R426D mutant localize to replication foci containing PCNA. Previous research from several laboratories has documented stereotypical patterns of PCNA foci that are characteristic of the different stages of S-phase [29,30,31].

Cells in early S-phase show numerous small PCNA foci distributed throughout the nucleus. These foci can be resolved from each other using super-resolution microscopy but are difficult to resolve from each other by confocal microscopy [31]. This makes protein localization to early S-phase PCNA foci difficult to assess using conventional tools for microscopy. Mid S-phase is the stage when perinuclear and peri-nucleolar chromatin are replicated. The corresponding PCNA foci are relatively large and distant from each other, making co-localization of a protein of interest with PCNA straightforward. Late S-phase is the stage when pericentric heterochromatin replicates. In mouse cells, this stage shows striking patterns of large and well-resolved PCNA foci, but, because CAF-1 contains both a PIP and a PXVXL motif for binding to Heterochromatin Protein 1 (HP1) [32,33], CAF-1 p150 localization to late S-phase PCNA foci persists even if the PIP is mutated because p150 retains the ability to bind to HP1 [18]. Because of these constraints, we focused our attention on cells with patterns of PCNA foci characteristic of mid S-phase. As reported in previous studies [4,34], we observed a distinct immunostaining pattern for PCNA with the monoclonal antibody PC10. During mid S-phase, staining of PCNA with PC10 occurred within foci located around the periphery of the nuclear envelope (Figure 4B, top panel, PC10). However, these cells also showed numerous PCNA foci around the periphery of nucleoli as well as many small foci within the interior of the nucleus (Figure 4B, top panel, PC10). Asynchronously proliferating mouse NIH 3T3 cells were transduced with lentiviral vectors, and expression of wild-type and mutant forms of GFP-p150 was induced with doxycycline. Cells were fixed and permeabilized for PCNA detection with the PC10 monoclonal antibody. Mid-S phase cells were identified based upon their stereotypical pattern of PCNA staining at the nuclear periphery. In this subset of S-phase cells, GFP-p150 co-localized with most, and perhaps all, of the PCNA foci located at the nuclear periphery (Figure 4B, upper panel). In mid S-phase cells, wild-type GFP-p150 was not present in the large foci of heterochromatin that can be detected by DAPI staining of mouse cells (Figure 4B, third row, upper panel). In striking contrast, GFP-p150 R426D did not co-localize with PCNA foci at the nuclear periphery and was anomalously found largely within the nuclear interior and the large heterochromatic foci (Figure 4B, lower panel). The Pearson’s correlation coefficient of GFP WT-mp150 was 0.6, and 0.25 for the mutant GFP-p150 R426D. This is consistent with previous observations [18], that when CAF-1 is unable to bind to PCNA, it prematurely associates with pericentric heterochromatin in early and mid S-phase.

#### 2.2.3. Arginine 426 Is Important for DNA Replication-Dependent Nucleosome Assembly

We next sought to compare the activities of CAF-1 p150, wild-type, and the R426D mutant in a replication-coupled chromatin assembly assay [16,35,36]. As a first step to performing this assay, we expressed CAF-1 p150 wild-type and the R426D mutant in a rabbit reticulocyte lysate coupled transcription and translation system and labelled the proteins with [^35^S]-methionine. The wild-type and R426D mutant proteins were present at comparable levels (Figure 4C, left panel). In the replication coupled chromatin assembly system, an S100 extract from cultured human HEK-293 cells provides all the cellular DNA replication enzymes and histones but lacks CAF-1 p150 [37]. To this extract, a plasmid containing an SV40 origin of replication and the viral replication protein, SV40 large T antigen, were added to support SV40 DNA replication. Radiolabeled dATP was also added to visualize the replication products by autoradiography.

In the absence of nucleosome assembly, the plasmid is replicated and radiolabeled, and the products of replication are topologically relaxed DNA molecules, demonstrating a lack of histone loading. In contrast, when the recombinant CAF1 protein (p150, p60, p48) or the p150 subunit alone are added, they bind to histones in the S100 extract and promote nucleosome assembly coupled with DNA replication. DNA topoisomerase enzyme I, which is present in the S100 extract (and further supplemented in these assays to ensure sufficient levels), removes the positive supercoiling in the linker DNA that connects nucleosomes, leaving negative supercoiling due to DNA wrapping around histone octamers (evident upon subsequent removal of the histones and electrophoresis). Thus, efficient nucleosome assembly in the presence of CAF-1 results in the formation of plasmid DNA that is replicated (hence radiolabeled) and negatively supercoiled (as indicated in Figure 4C), thus highlighting the interdependency of chromatin assembly, plasmid supercoiling and DNA replication.

Equal amounts of CAF-1 p150 wild-type and R426D mutant were added to the chromatin assembly assay system. After replication and chromatin assembly reactions were completed, the DNA was purified and analyzed by agarose gel electrophoresis. Gels were stained with ethidium bromide to visualize total DNA, then dried and subjected to autoradiography to specifically observe the replicated DNA. In the absence of recombinant CAF-1, replicated DNA was not extensively supercoiled (Figure 4C, rightmost panel, lane 1). However, when recombinant CAF-1 was added, the replicated DNA was selectively supercoiled (Figure 4C, rightmost panel, lane 2, compared to middle panel, lane 2), indicative of nucleosome assembly. Similarly, when increasing amounts of the in vitro translated wild-type CAF-1 p150 were added, the replication products were supercoiled (Figure 4C, rightmost panel, lanes 3–5). In striking contrast, when increasing amounts of the R426D mutant were added, much less supercoiling was observed (Figure 4C, rightmost panel, lanes 6–8). This indicates that the CAF-1 R426D mutant is deficient in nucleosome assembly. (Note that residual CAF-1 levels present in S100 extracts after immunodepletion resulted in low levels of supercoiling, as observed in the negative control, Figure 4C, rightmost panel, lane 1, and to a similar degree in other lanes).

### 2.3. A Predicted Long α-Helix of CAF-1 p150 Abruptly Ends with the PIP

Earlier studies have shown that the KER domain of p150 is essential for nucleosome assembly by CAF-1 [16]. However, it was not clear how the KER domain, named because of its high content of lysine, glutamic acid, and arginine residues, exerted its role in CAF-1-mediated nucleosome assembly. We were interested in domains adjacent to the CAF-1 p150 PIP that might modulate its affinity for PCNA. Interestingly, several secondary structure algorithms predict the presence of a very long α-helix comprising the KER domain. The prediction for human CAF-1 p150 using the PSIPRED algorithm from the UCL Department of Computer Science Bioinformatics group was for a 119-residue helix from Thr330 to Phe448 (Figure 5) [38]. At 3.6 residues per turn, this would represent a 33-turn α-helix. We applied the secondary structure prediction algorithms to p150 homologues from several species, ranging from yeast to humans. We made two striking observations. First, despite species-specific variations in helix length (100–120 amino acids; Appendix A), each species was predicted to have a very long α-helix. Second, it was most striking that in every species the long α-helix was predicted to terminate after the first aromatic residue of the PIP (Phe448 in Figure 1B and Figure 5, CAF-1 p150 PIPs). In other words, seven of the eight residues that constitute the PIP were predicted to be part of the helix in each species. In many, but not all species, the C-terminal end of the predicted helix was dictated by the presence of a helix-breaking proline a few residues after the PIP (Pro452 in Figure 5). 

Analysis of CHAF1A p150 by AlphaFold2 [39,40] predicts a long alpha helix covering residues 335–448, matching almost exactly the predicted helix for residues 330–448 with confidence (average per-residue confidence score (pLDDT) score of 79.2). Furthermore, we used AlphaFold-Multimer to evaluate the possibility that the KER domain does indeed form a coiled-coil, as predicted (Figure 6C).

An unusually long α-helix (119 amino acids) is predicted to end abruptly in the PIP RFF residues. The CAF-1 p150 PIP is surrounded by a red rectangle. To address this surprising prediction, CAF-1 p150 fragments that contained the predicted long α-helix including the PIP were expressed in *E.*
*coli*, purified, and their biochemical and biophysical properties were characterized. This fragment of human p150 (Figure 5, residues 342–474) was expressed as a His-tagged fusion protein, hereafter referred to as CAF-1 p150L. Despite purification in the presence of 2.5 M NaCl, the CAF-1 p150L protein co-purified with nucleic acid contamination. Further treatment with benzonase and three sequential 1 M NaCl washes were required to remove the contamination.

#### 2.3.1. CAF-1 p150L Is Highly Helical

Circular dichroism (CD) spectroscopy was used to determine whether purified CAF-1 p150L protein forms a long α-helix as predicted. The CD spectrum of CAF-1 p150L clearly showed features typical of a mainly helical protein (Figure 6A, upper panel), as shown by the strong negative molar ellipticity at 222 nm and 208 nm.

#### 2.3.2. CAF-1 p150L Is Monomeric 

Recent studies from Panne et al. [41] showed that the KER domain of *S. cerevisiae* Cac1 (the yeast homologue of human p150) is the principal DNA-binding domain of the CAF-1 protein. The KER domain is also predicted to be a coiled-coil domain, where two or more α-helices coil around each other [42,43,44]. Alpha helices that form short coiled-coils (30–40 residues) are relatively common in eukaryotic transcription factors that form DNA-binding domains known as leucine zippers [45]. Proteins that form long coiled-coil domains have thus far only been identified in cytoskeletal motor protein families such as myosin, kinesin, and dynein [46,47]. Most of these proteins contain a repetitive pattern of seven amino acids (heptad) that functions to bury their hydrophobic surfaces between hydrophilic amino acids. No heptad repeat is present in the CAF-1 p150 KER domain. 

Nonetheless, to investigate the possibility that p150L might form a coiled-coil, size exclusion chromatography (SEC) coupled to multi-angle light scattering (MALS) was used. We observed a principal peak at ~14.2 mL and a minor peak at ~12.0 mL (Figure 6B, blue trace). MALS determined that the principal peak was ~19 kDa, corresponding to monomeric p150L (theoretical MW ~17 kDa). It is also noteworthy to mention that the SEC elution profile of CAF-1 p150L was not typical of a globular protein. For instance, the elution profile of BSA (Figure 6B, red trace) showed two peaks, corresponding to a dimer (~134 kDa) and a monomer (~67 kDa). Furthermore, the minor dimeric p150L peak elutes at approximately the same volume as dimeric BSA, which is drastically earlier than monomeric p150L. Since a coiled-coil dimeric arrangement would have similar hydrodynamic properties as a single long α-helix, the minor peak is most likely representing an elongated dimeric arrangement. Indeed, such an arrangement is predicted by AlphaFold-Multimer [39,40] for the p150L protein (Figure 6C).

#### 2.3.3. CAF-1 p150L Adopts an Elongated Rod-like Structure 

Size exclusion chromatography coupled to small-angle X-ray scattering (SEC-SAXS) was used to evaluate the approximate shape of p150L. An average scattering profile of all frames within the SEC elution peak corresponding to the p150L monomer was calculated (Figure 7A). The particle distribution function P(r) curve was obtained after raw data processing using the pair distribution function GNOM (Figure 7B). Dummy atom modeling of the SAXS results suggests that the protein adopts an elongated form with a shape related to that of “beads on a string”, with 5 or 6 spherical beads, each with a diameter of 20–25 Å (Figure 7C). This would roughly amount to 4–5 α-helical turns per “bead” (the pitch of an alpha helix is ~5.4 Å), or 13–16 residues (3.6 residues per α-helical turn), considering that the secondary structure is mostly α-helical in nature which is consistent with the CD data (Figure 6A). The overall length of the particle (D max) is in the range of 170–180 Å (Rg ~ 45–50 Å).

#### 2.3.4. CAF-1 p150L May Bind Directly to PCNA

Secondary structure prediction algorithms included the PIP residues observed in the crystal model, except the last aromatic residue of RFF, within the last turn of the predicted long helix. Given this, it was not clear that p150L would bind PCNA when held in an α-helical conformation. We attempted to determine whether p150L was bound directly to PCNA by performing nuclear magnetic resonance (NMR) and fluorescence quenching experiments. 

The appearance of the 1H-15N BEST HSQC for 15N-labelled p150L alone suggested that p150L was not a well-folded, globular protein with a distinct three-dimensional structure. The visible signals are poorly dispersed and reduced in number (we should see ~139 signals for a well-folded p150L protein). Overall, the spectrum agrees with SEC and SAXS data, with broad and non-visible peaks likely a consequence of multiple conformations on the intermediate timescale, although further analysis is required to confirm this. The PCNA homo-trimer (88.7 kDa) is a relatively large protein from an NMR standpoint. PCNA is sufficiently large that it tumbles slowly in solution, which, in turn, results in very few NMR peaks being detected. When we mixed CAF-1 p150L with PCNA, some of the well-resolved peaks corresponding to CAF-1 p150L disappeared or were significantly shifted, which is observed clearly from the overlay (Figure 8). This is consistent with CAF-1 p150L binding to PCNA, thus generating a complex (minimal size of p150L + PCNA homotrimer = 107 kDa) with slow tumbling time, which results in a loss of signal. The strong visible peaks, many of which undergo chemical shift perturbations, are most likely in regions of p150L that are unfolded. 

Overlay of NMR spectra from 15N-labelled p150L and a complex of p150L with PCNA. 1H-15N HSQC NMR spectrum of 15N-labelled p150L is shown in black. The signals are not properly dispersed, which is not characteristic of a properly folded globular protein with a single conformation. Upon addition of a molar excess of PCNA monomer (spectrum shown in blue), most of the p150L spectrum disappears, consistent with p150L binding to a large protein such as PCNA. 

To test the hypothesis that there is an interaction between p150L and PCNA, we performed a complementary experiment, namely fluorescence quenching, which exploits the inherent absence of tryptophan residues in CAF-1 p150L and the presence of the single Trp residue per PCNA monomer. The intrinsic fluorescence of the PCNA signal is shown in the absence or presence of a four-fold molar excess of p150L over PCNA monomer (Figure 9, red and black traces respectively). As expected, the emission spectrum of CAF-1 p150L alone is very small due to the absence of tryptophan residues (green trace; Figure 9). 

Clearly, the addition of CAF-1 p150L to PCNA quenched a small portion of the tryptophan signal derived from PCNA, which is consistent with complex formation (Figure 9). However, the fact that only a small portion of the PCNA-derived fluorescence emission was quenched upon the addition of a four-fold molar excess of p150L suggests weak binding under these solution conditions. Although the NMR and fluorescence quenching experiments support that p150L directly binds to PCNA, further experiments are needed to determine the binding affinity. 

## 3. Discussion

In this manuscript, we have shown that the PIP of CAF-1 binds to PCNA in a manner that is unprecedented among the numerous known structures of PIPs bound to PCNA. Among other features, Arg 426 is involved in forming a cation-π interaction with the aromatic ring of Tyr250 from PCNA (Figure 2 and Figure 3). Because Arg426 is not conserved among other PCNA-binding proteins, the cation-π interaction can thus far be only formed by CAF-1 homologues from a number of species, with the exceptions of *S. cerevisiae* or *S. pombe* (Figure 1B). CAF-1 sequences from multiple yeast species are canonical with a conserved glutamine in the first position, as in *S. Cerevisiae* it is **Q**SR**I**GN**FF**. The PIP box sequences of CAF-1 analyzed in multiple Candida species are also found to be canonical (Appendix A). This could partly be because the binding mechanism involving CAF-1 in the entire fungal species might be regulated differently, and future studies are necessary to further understand this. 

Similarly, the CAF-1 PIP box sequence of Msh3, which is a part of the DNA mismatch repair machinery, is 19-APAR**Q**AV**L**SR**FF**QSTGSLKSTSSST-41 and in accordance with the canonical sequence, the first PIP box residue corresponds to a glutamine, whereas in contrast, in the PIP box of CAF-1 this is replaced by a lysine at this position, thus making the binding mode of CAF-1 p150 non-canonical and unique in the case of human CAF-1 sequence 442-**K**AE**I**TR**FF**-449 and mouse CAF-1 sequence 421-**K**AE**I**TR**FF**-428. Previous research from our lab has also shown that substitution of the CAF-1 p150 lysine by a glutamine to mimic a canonical PIP-box sequence enhances the direct binding of CAF-1 p150 PIP to PCNA at moderate ionic strength [18]. In addition, a synthetic peptide of p150 with lysine to glutamine substitution could also deplete PCNA from S100 extracts. This signifies the importance of having a non-canonical lysine at the starting position of the CAF-1 PIP box to avoid its non-specific binding to PCNA and the presence of an arginine residue that mediates a unique and specific cation-pi interaction with the tyrosine 250 of PCNA, as we observed in the crystal model. In vitro, the Arg426 of CAF-1 p150 is required for PIP binding to PCNA and DNA replication coupled nucleosome assembly (Figure 4A,C), indicating that this interaction is physiologically important. In vivo, Arg426 is needed for targeting CAF-1 p150 to PCNA-containing DNA replication foci during mid S-phase (Figure 4B). 

Recent studies have also reported specialized PIP boxes that target PCNA partners for proteasomal degradation known as PIP degrons [48,49]. The motif for PIP-degron is QXX_Ψ_TD_θθ_XXXB. The five most striking features of a PIP degron are a conserved glutamine at position 1(Q), an aliphatic residue _Ψ_ at position 4 (leucine, isoleucine, valine, or methionine), TD residues that give higher PCNA binding specificity, two aromatic residues _θθ_ at positions 7 and 8 (phenylalanine, tyrosine, or tryptophan) and a basic residue (either lysine or arginine) as the 4th residue after the two aromatic residues. Studies have highlighted that removal of PIP degrons are also important for facilitation of DNA repair mechanisms and translesion synthesis [49]. Even though, arginine (R) to aspartic acid (D) mutation in the CAF-1 p150 PIP (**K**AE**ITDFF**QKP**K**T) resembles some residues in the PIP degron, it is unlikely that this sequence would be recognized for a PIP-degron mediated degradation due to the absence of highly conserved glutamine in the PIP degron motif.

Within the context of the full-length p150 protein; the CAF-1 PIP is predicted to occur at the C-terminal end of a very long α-helix (100–120 residues depending on the species) that encompasses the entire KER domain (Figure 5 and Figure 6A), which is one of two DNA-binding domains within CAF-1 p150. Previous studies have demonstrated that the KER domain is reported to have DNA binding activity (40bp) [41]. We also observed that the highly charged KER domain of CAF-1 tightly binds to DNA and high salt washes combined with benzonase digestion were required to eliminate the bound DNA. In solution, the long α-helix that spans the KER domain and the PIP analyzed by SEC-MALS has been shown to be capable of predominantly forming monomers and a small proportion of dimers (Figure 6B). Several studies also indicate that the purified shorter versions of CAF-1 p150 and cac1 tend to oligomerize (dimerize) in the absence of their binding partners [7,50,51,52]. The AlphaFold2 model analysis predicts the presence of a long alpha helix covering residues 335–448, matching almost exactly the PSIPRED predicted helix for residues 330–448 we observed in our analyses. (Figure 5 and Figure 6C). SEC-SAXS analysis shows that CAF-1 p150L adopts an oblong shape (Figure 7C). Although prediction algorithms claim that this helix will become a coiled-coil, AlphaFold2 multimer analysis predicts that the p150L construct does not form a typical coiled-coil, at least not with itself. A small section of 17 amino acids (Arg432-Phe449) may mediate dimerization (Figure 6C), and this would lead to a longer species than a typical coiled-coil. Furthermore, the dimerization of the p150 subunit observed in previously reported studies is mediated by a region of 36 amino acids located between residues 642–678 outside of the KER domain [7]. 

The crystal model with a 20 amino acid peptide of CAF-1 with PCNA did not show the presence of an apparent N-terminal helix or any other structured residues other than the three amino acids—RFF. This could probably be due to an absence of having enough amino acids to form a helix within the peptide. However, secondary structure predictions using PSIPRED, in silico modeling using AlphaFold2 and in solution SEC-SAXS data demonstrate that the p150L domain of CAF1 adopts a long alpha-helical structure. However, despite its presence at the end of a long α-helix, we provide preliminary evidence suggesting that the juxtaposition of the KER domain and the PIP is not an irremediable obstacle to PCNA binding (Figure 8 and Figure 9). Nevertheless, it cannot be excluded that upon binding to PCNA, part of this KER domain, including the PIP box, rearranges in order to exhibit its residues involved in the binding interface. In other words, it is possible that binding of CAF-1 p150L to PCNA would stabilize the N-terminal region of this structure as a sort of higher-order regulation, probably involving multiple post-translational modifications on either of these proteins. 

***Is there a PCNA ring dedicated to nucleosome assembly?*** One argument that is often invoked to explain how functionally distinct enzymes (e.g., Pol δ, FEN1, and DNA ligase during lagging strand synthesis) might exert their activities by binding to the same PCNA ring is the sequential binding and dissociation of each enzyme. This is often invoked in cases where the binding of one or more of the enzymes is mutually exclusive because of steric hindrance. For instance, this is the case for Pol δ and DNA ligase I binding to PCNA that occludes the entire front face of the homo-trimer [53,54]. However, we feel that sequential binding and dissociation of each enzyme is not a viable model for two reasons. This is best illustrated by the roles of Pol δ, FEN1 and DNA ligase I in lagging strand synthesis. The first argument is that any need for Pol δ to dissociate from a single PCNA ring, in order to allow access to FEN1 and DNA ligase I, simply defeats the role of PCNA as a processivity factor for Pol δ. A distributive (binding and dissociation) model for access to PCNA implies that the affinities of each enzyme for PCNA are sufficiently different to ensure that the enzymes act according to a specific sequence (e.g., Pol δ must gain access to PCNA first). This model is flawed for an obvious reason. Assuming that Pol δ has the highest affinity for PCNA, there is no apparent means to prevent it from competing with the enzymes that need to act on DNA synthesized by Pol δ, namely FEN1 and DNA ligase I. 

A more plausible, but unproven, option than the “binding and dissociation” model is the “multiple functionally dedicated rings” model. In this case, multiple PCNA rings are loaded sequentially by RF-C and, in the simplest of cases, each PCNA ring becomes functionally specialized by associating with a unique enzyme according to a specific sequence (e.g., Pol δ first, then FEN1 and DNA ligase I last). Once the appropriate enzyme has bound to the first ring (e.g., Pol δ), functionally dedicating that PCNA ring to DNA synthesis could involve several mechanisms (e.g., ring-specific modifications of PCNA and/or the PCNA-binding enzyme), but in many circumstances, may involve steric occlusion of that PCNA ring from functionally unrelated PCNA-binding enzymes. This is clearly the case for both Pol δ and DNA ligase I because their binding to PCNA completely occludes the front face of PCNA to which functionally different PCNA-binding enzymes might otherwise have access [53,54]. 

It is noteworthy that none of the other canonical DNA replication proteins that we looked at, including p21, FEN1, DNMT1, and DNA ligase I, exhibit the presence of long alpha helices that terminate in their PIPs. (Appendix A). Similarly, the error-prone DNA polymerases Pol kappa and Pol iota, which contain non-canonical PIPs, also lack a predicted long a-helix adjacent to their PIPs (Appendix A). Thus far, the model that we identified in CAF-1 p150 is unique, which might have functional significance for regulating CAF-1 binding to DNA and PCNA and, ultimately, chromatin assembly.

***Is the long α-helix of CAF-1 p150 acting as a DNA binder, a DNA bender, or a DNA ruler?*** In many species, ranging from yeast to humans, the KER domain is predicted to form an unusually long α-helix of CAF-1 p150 and these secondary structure predictions were borne out by our CD experiments with the human p150L protein. A striking feature of the predictions is that, regardless of the species, the long α-helix always ends at the first aromatic residue of the CAF-1 p150 PIP. Because the KER domain of *S. cerevisiae* Cac1 (the orthologue of human p150) has been firmly established as one of the two DNA-binding domains of budding yeast CAF-1 [41], the juxtaposition of the KER DNA-binding domain and the PIP within the same α-helix suggests that DNA and PCNA binding are coordinated. Why might that be the case? The first possibility is that this may be part of a mechanism to ensure that the nascent DNA immediately behind PCNA is free of protein impediments that might interfere with the formation of (H3-H4)_2_ tetramers on nascent DNA. In this scenario, the KER α-helix would merely serve as a DNA binder that is present immediately behind the PCNA ring to which CAF-1 is associated, which must act near the point where nascent DNA emerges from the replicative polymerases [55,56]. Robust DNA binding by the KER α-helix would occlude the freshly synthesized DNA from irrelevant DNA-binding proteins and keep the DNA substrate pristine for deposition of H3-H4 by CAF-1. The second possibility stems from the fact that 147 bp of DNA must be wrapped into 1.65 left-handed super helical turns around histone octamers to form nucleosome core particles [57]. It turns out that B-form DNA is one of the stiffest known polymers, and its persistence length, the distance below which a polymer is essentially a rigid rod, is 150 bp in 0.1 M NaCl. This is a lot longer than the 73 bp needed to accommodate the first intermediate in nucleosome assembly, namely the deposition of the (H3-H4)_2_ tetramer onto nascent DNA [55,58]. The rigidity of DNA against bending stems from electrostatic repulsion among the phosphate groups that must necessarily come into close proximity as a result of DNA bending. It therefore stands to reason that charge neutralization of the phosphate groups on the sides of the double helix by a DNA binding protein facilitates deformation (bending) of the DNA double helix. Given this, it seems possible that the function of the KER α-helix may be to prepare the nascent DNA for H3-H4 deposition by bending the DNA substrate.

There exist other lines of evidence supporting the latter model, that the α-helix may be acting as a DNA binder and/or a DNA ruler to assist in appropriate loading and wrapping of DNA around histones. Several lines of evidence argue that new (H3-H4)_2_ tetramers are assembled from two CAF-1 complexes that each bind to an H3-H4 dimer [41]. A sufficient length of nascent DNA must be available before CAF-1 deposits two H3-H4 dimers onto DNA to form (H3-H4)_2_ tetramers. One can readily calculate the maximal length of DNA that can be covered by the KER α-helix, namely 119 residues of a prototypical Corey-Pauling α-helix (3.6 residues/turn; 5.4 Å/turn for the pitch of the helix). This adds up to 178 Å, which is equivalent to approximately 52 bp of B-form DNA (3.4 Å/bp). Fifty-two base pairs are too short to accommodate the 73 bp needed to assemble (H3-H4)_2_ tetramers onto DNA. However, because two CAF-1 complexes are needed to assemble (H3-H4)_2_ tetramers, the maximal amount of DNA contacted by two KER α-helices is 104bp (2 × 52 bp), which is sufficient to accommodate a single (H3-H4)_2_ tetramer onto nascent DNA. In this scenario, the KER α-helices of two CAF-1 complexes would act in a concerted fashion and serve as “*DNA rulers*” to ensure that H3-H4 are not deposited onto DNA before a sufficient length of freshly synthesized DNA has emerged from the replisome to accommodate the formation of (H3-H4)_2_ tetramers. This *DNA ruler* function of the KER α-helices is not mutually exclusive with the aforementioned *DNA binder* and *DNA bender* functions. Lastly, we note that the fact that the KER DNA-binding domain and the PIP are part of the same α-helix implies that, once the DNA-binding domain is engaged on the DNA substrate, the position of the PCNA ring with respect to DNA is likely fixed, and the PCNA ring to which CAF-1 is bound may no longer slide freely along the DNA. This bookend function of the KER DNA-binding domain and PCNA would contribute to ensuring that (H3-H4)_2_ tetramers are uniformly spaced along the DNA.

In conclusion, the experiments presented here have raised many fascinating issues for future research. In particular, the juxtaposition of the KER DNA-binding domain and the PIP raises the possibility that DNA binding and PCNA binding may be coordinated to bring about efficient H3-H4 deposition onto nascent DNA without functional interference from the numerous other enzymes that need to bind PCNA at replication forks. 

## 4. Materials and Methods

### 4.1. Constructs for Expression of Mouse CAF-1 p150 in Rabbit Reticulocyte Lysate

A cDNA fragment for expression of wild-type mouse CAF-1 p150 was inserted between the *Not*I and *Bam*HI/*Bgl*II sites of the pCITE-4a^+^ vector (Novagen) [32]. The PIP sequences in humans and mice are identical: 421-KAEITRFF-428 (mouse numbering). Using site-directed mutagenesis, the following mutations in the PIP region of the full-length mouse p150 (CHAF1A) protein were generated: pCITE4a p150 R426D, pCITE4a p150 R426S, and pCITE4a p150 R426L. Wild-type CAF-1 p150 and PIP variants were expressed following the manufacturer’s instructions in the rabbit reticulocyte lysate using the T_N_T quick system for coupled transcription and translation (L1170-Promega). The expressed proteins were labeled with [^35^S]-methionine and subsequently used for nucleosome assembly assays.

### 4.2. Generation of Lentiviral Vectors for Doxycycline-Inducible Expression of GFP CAF-1 p150

To study the localization of GFP CAF-1 p150 wild-type and mutants relative to that of PCNA, cDNAs encoding mouse p150 were cloned into a doxycycline-inducible lentiviral vector pCW-Cas9 plasmid encoding a Tet_ON_ (minimal CMV) promoter (Addgene plasmid 50661). The Cas9 insert was replaced by wild-type or mutant mouse CAF-1 p150 using the restriction enzymes *Nhe*I and *Bam*HI. GFP-p150 wild-type and mutant fusion proteins were excised from the EGFP-C1 vector (Clonetech) using Nhe1 and BamH1 restriction enzymes as previously reported [18]. *Nhe*I and *Bam*HI digested fragments were run on 0.8% agarose gel and the corresponding DNA bands: vector—pCW-Cas9 (7036 bp) and inserts—wild-type mp150, R426D (3983 bp) were gel purified using Qiagen DNA extraction spin prep columns. Purified DNAs were ligated using the Takara DNA ligase enzyme, following the manufacturer’s recommended protocol. Positive clones were selected on Luria-Bertani medium containing 100 µg/mL ampicillin, and after DNA isolation, the clones were verified by DNA sequencing. 

### 4.3. Cell Culture and Transfection

HEK-293T and NIH3T3 cells were cultured in Dulbecco’s modified Eagle medium (DMEM) with 10% foetal bovine serum without antibiotics. HEK-293T cells were used for co-transfection of the lentiviral packaging plasmids and gene of interest plasmids for viral packaging, whereas NIH3T3 cells were used for transduction of the individual viruses and subsequent immunofluorescence analysis. HEK-293T cells, seeded in 100 mm tissue culture dishes at 80% confluency, were transfected with psPAX2 and pMD2G along with plasmids expressing wild-type and mutant mp150 using TransIT—Lenti transfection reagent (Mirus) as per the manufacturer’s recommendations. Lentiviral supernatants were harvested at 48 h and 72 h after transfection, filtered using a 0.45-micron filter to remove the cell debris, and used to transduce target NIH3T3 cells seeded on glass-bottom 35 mm dishes coated with poly-L-lysine. Polybrene was added at a final concentration of 8 µg/mL to the viral supernatant to aid viral transduction. After 24 h of transduction, viral supernatant containing media was removed from NIH3T3 cells and fresh media containing doxycycline at a final concentration of 2 µg/mL was added to the cells. Turbo GFP was used as a positive control for monitoring transduction efficiency and the cells were confirmed to be free of mycoplasma using the Mycoalert mycoplasma detection kit (Lonza, Basel, Switzerland).

### 4.4. Immunofluorescence Detection of DNA Replication Foci in S-Phase Cells

NIH3T3 cells seeded on glass bottom 35 mm dishes were fixed with 2% paraformaldehyde in PBS at room temperature for 15 min. The cells were then treated with −20 °C methanol to expose the PC10 epitope. After fixation, the cells were washed three times with 1X PBS containing 1% BSA. PC10 primary monoclonal antibody was added to the cells at a 1:80 dilution and incubated for 2 h at room temperature. Cells were washed three times with 1X PBS containing 1% BSA to remove unbound primary antibody and incubated with secondary goat anti-mouse IgG antibody at 1:400 dilution for 2 h at room temperature. After three washes in 1X PBS containing 1% BSA; 4′,6-diamidino-2-phenylindole (DAPI) stain was added at 0.1 µg/mL to the cells and the images were acquired as a Z stack on a Zeiss LSM 880, an inverted confocal microscope using a plan apochromat 40X/1.4 NA oil immersion objective (scale bars = 10 µm). The single images of the Z stacks were analyzed and merged using the proprietary ZEN software from Zeiss (Oberkochen, Germany).

### 4.5. Crystallography of PIP Bound to PCNA

A CAF-1 p150 synthetic peptide with the sequence IKAEKAEITRFFQKPKTPQA was synthesized at the University of Cambridge in-house protein and nucleic acid facility (PNAC). For use in crystallization trials, the synthetic peptide was dissolved in demineralized water to a concentration of 23 mM. Initial crystallization conditions were further refined and optimized with the help of the PACT suite (Qiagen) and Additive Screen (Hampton Research), which contain multiple reagent libraries that enhance the solubility and crystallization of biological macromolecules. Crystal yield was greatly improved under these conditions, and we obtained crystals that exhibited flat surfaces and sharp edges indicating regular lattice plane formation. Well conditions were 0.1M MMT buffer pH 5, 22% (w/v) PEG 1500, and 1.0 M Guanidine hydrochloride. These crystals were subsequently used for diffraction studies. The diffraction data was collected by Drs. Tomasso Moschetti and Andrew Thompson at the Proxima 1 beamline of the SOLEIL synchrotron (Saint-Aubin, France). 

### 4.6. Isothermal Titration Calorimetry (ITC)

Recombinant PCNA, PCNA-Y250I, and the PIP peptides (wild-type, R426D) were expressed and purified as described in the Appendix A—“Expression and purification of recombinant proteins”. The proteins were dialyzed against three 4-litre changes of ITC buffer containing 10 mM sodium phosphate, pH 7.0, and 10 mM NaCl at 4 °C overnight. The concentrations of PCNA and PIP peptides were determined by measuring tyrosine absorption at A_280nm_. ITC titrations of peptides into PCNA were performed at 23 °C in the ITC buffer (10 mM sodium phosphate, pH 7.0, 10 mM NaCl) using a MicroCal VP-ITC system. The concentrations of injected wild-type and mutant PIP in the syringe varied from 200 to 500 μM, whereas PCNA in the ITC sample cell varied depending on the experiment from 20 to 50 μM PCNA monomer. The protein and peptide samples were degassed before the ITC run. Data analysis was performed using the MicroCal Origin 7.0 software (OriginLab Corporation, Northampton, MA, USA) and all experiments fit the single binding site model. 

### 4.7. Replication-Dependent Nucleosome Assembly Assays

Replication reactions were assembled as previously reported [35,36]. The chromatin assembly reactions contained a final concentration of 40 mM Hepes-KOH, pH 7.5, 8 mM MgCl_2_, 0.5 mM dithiothreitol, 0.2 mM each of dTTP, dCTP, dGTP, 25 µM of dATP, 3 mM ATP, 0.1 mM each of TTP, CTP, GTP, 25 µM [α-32 P] dATP (3000 Ci/mmol; 10 µCi/µL), 40 mM creatine phosphate, 0.05 units/µL creatine phosphokinase, and 100 ng of pSV011^+^ origin containing plasmid, 25 ng/µL of T-antigen (T-antigen was a kind gift from Dr. Bruce Stillman at Cold Spring Harbor Laboratory), 0.75 units/µL Topoisomerase I, 0.16 units/µL Topoisomerase II.

Each 12.5 µL reaction had purified full-length recombinant CAF-1 as a positive control for nucleosome assembly reactions. The experimental tubes contained in vitro translated [^35^S]-methionine labelled either with wild-type p150 or p150 R426D in increasing amounts. The replication reactions were incubated for 45 min at 37 °C before being restarted for another 45 min after the addition of 0.25 µL of 1 mg/mL purified H2A.H2B dimers. After incubation, the reaction was stopped by the addition of a stop solution containing 10 mM EDTA and 0.5% SDS. The reaction mixture was first digested with ribonuclease A (20 µg/mL) for 15 min at 37 °C and then with 1 mg/mL of pronase for 1 h at 37 °C. 

The reactions were extracted using phenol, chloroform, and isoamyl alcohol (25:24:1) and the DNA was then precipitated with sodium acetate and ethanol. The pellet was rinsed with 70% ethanol and dissolved in 15 µL of 1X TE buffer, pH 8.0, and mixed with 3 µL of 6X bromophenol blue sample buffer. The products of replication were analyzed using a native 1.25% agarose gel in 1X TAE buffer with 1 mM MgCl_2_, without the addition of ethidium bromide. After gel migration, the gel was stained with ethidium bromide for 10 min, destained, and imaged. The ethidium bromide-stained gel was fixed for 30 min in a fixation solution containing 10% acetic acid and 10% methanol. The gel was rinsed once in water to remove the acid and dried using a gel dryer with Whatman filter papers and covered in saran wrap, without heat for 30 min, followed by heating at 70 °C for an additional 30 min. After the gel was completely dry, it was transferred to an autoradiography cassette with an imaging plate for 1–2 days. The gel was subsequently imaged and analyzed using a phosphor imager (Typhoon FLA 7000) to visualize the replicated DNA.

### 4.8. Circular Dichroism (CD) 

Purified CAF-1 p150L protein was dialyzed overnight against 10 mM sodium phosphate buffer, pH 7.2, 10 mM NaCl, and 0.1 mM TCEP. Due to the absence of tyrosine and tryptophan residues in the CAF-1 p150L protein, the protein concentration was measured after dialysis using a spectrophotometer at A_205nm_ [59]. The protein was used in the CD experiment at a final concentration of 4.68 μM. Far-UV CD spectra were collected at room temperature using a 0.02 cm path length cuvette on a JASCO J810 CD spectropolarimeter. Buffer effects were removed by running a control spectrum on the dialysis buffer alone. CD spectra were acquired with a data pitch of 0.5 nm over a range of 260 nm to 190 nm, with a 1 nm bandwidth, a scan speed of 20 nm/min, a response time of 4 s, and 4 scans per spectrum. Relative ellipticity was converted to mean residue molar ellipticity according to published literature (Fasman, 1996).

### 4.9. Intrinsic Tryptophan Fluorescence Spectroscopy

Fluorescence experiments were performed on an Agilent Cary Eclipse fluorimeter at room temperature. Buffer conditions were identical for all protein components (10 mM sodium phosphate, pH 7.2, 10 mM NaCl, 0.1 mM TCEP). Samples were excited at A_280nm_ with a 5 nm bandwidth and the emission spectra between 295 nm and 420 nm were collected. Measurements were made for 5 μM PCNA monomer (1 tryptophan residue), 20 μM CAF-1 p150L (no tryptophan and tyrosine residues), and a mixture containing 5 μM PCNA monomer and 20 μM CAF-1 p150L. Both samples were extensively dialyzed against the same buffer and prepared from stock samples to ensure identical conditions. 

### 4.10. Size Exclusion Chromatography Coupled to Multi-Angle Light Scattering (SEC-MALS)

SEC-MALS was used to determine the molecular mass and oligomerization status of CAF-1 p150L. Samples were injected onto a Superdex 200 Increase 10/300 GL column at 0.35 mL/min using an ÄKTAmicro FPLC (GE Healthcare). The running buffer contained 10 mM sodium phosphate, pH 7.2, 0.1 mM TCEP, and either 150 or 500 mM NaCl. Samples were passed through a Dawn HELEOS II MALS and OptiLab T-rEX online refractive index detectors (Wyatt Technology, Santa Barbara, CA, USA) after calibration with the BSA monomer. Data were processed with ASTRA Version 6.1.6.5 (Wyatt Technology, Santa Barbara, CA, USA). 

### 4.11. Size Exclusion Chromatography—Small Angle X ray Scattering (SEC-SAXS)

SEC-SAXS data were collected using an ÄKTAmicro FPLC and a Superdex 200 Increase 10/300 GL column (GE Healthcare), coupled inline to a BioXolver SAXS system (Xenocs) equipped with a MetalJet D2+ 70 kV X-ray source (Excillum) and a PILATUS3 R 300K detector (Dectris). Five hundred microliters of sample were injected into the SEC-SAXS system, and the chromatography was carried out at a flow rate of 0.05 mL/min. X-ray scattering images corresponding to 60 s of exposure were collected continuously at 20 °C during the elution, and an average scattering profile of all frames within the elution peak was calculated. Buffer scattering was then subtracted from the average scattering profile of the elution peak to yield the scattering curve of the sample. More data collection details are given in Appendix A.

### 4.12. Nuclear Magnetic Resonance (NMR)

NMR experiments were carried out on a 600 MHz Bruker Avance III spectrometer equipped with a 5 mm QCIP cryoprobe with Z gradients. All experiments were carried out at 293K. Two-dimensional ^1^H-^15^N BEST HSQC [60] experiments were carried out using the sequences downloaded from the NMRlib suite [61] of pulse sequences. Selective shaped pulses were used to excite a ^1^H sweep width of 4.5 ppm, centered at 8.75 ppm, with a recycle delay of 0.2 s. ^1^H and ^15^N sweep widths were set to 8417.509 Hz (14 ppm) and 1920.466 Hz (31.6 ppm) respectively, typically 240 t_1_ increments were acquired. NMR samples were prepared in 10 mM sodium phosphate buffer, pH 7.2, 10 mM NaCl, and 90% H_2_O/10% D_2_O. Uniformly ^15^N labelled CAF-1 p150L and unlabeled PCNA were prepared in identical buffers and added together from stock samples. Final sample concentrations were at 50 μM for CAF-1 p150L and 400 μM PCNA monomer. NMR spectra were processed with NMRpipe, a multidimensional spectral system based on UNIX pipes [62].

## Figures and Tables

**Figure 1 ijms-23-11099-f001:**
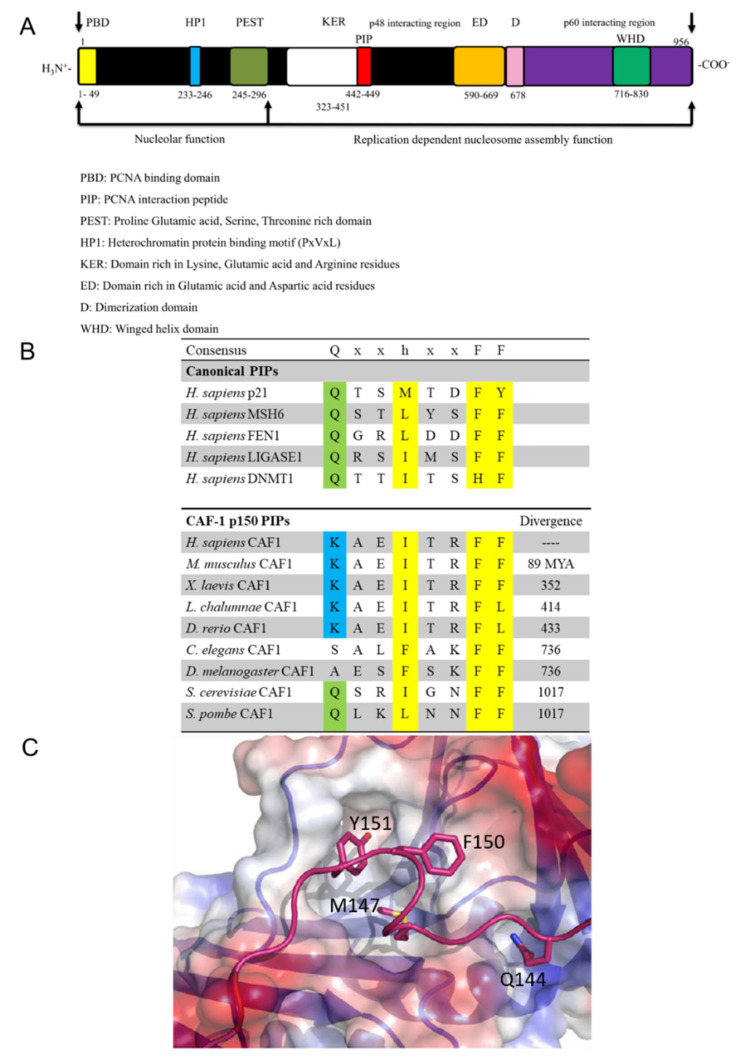
Differences between canonical PIPs and the non-canonical PIP in the p150 subunit of CAF-1. (**A**) Domains of the human CAF-1 p150 subunit (**B**) Amino acid alignment of canonical PIPs versus evolutionarily conserved CAF-1 p150 PIPs. The evolutionary divergence is shown in the extreme right column (million years ago = MYA) (**C**) Plug and socket binding mode in the structure of the canonical p21 PIP bound to PCNA [19]. The p21 PIP is shown in pink. PCNA hydrophobic surfaces are depicted in white, whereas positively and negatively charged surfaces are respectively in blue and red. Residues of the canonical PIP of p21 are shown in stick representation: Q144, M147, F150, and Y151. Q144 occupies the Q pocket of PCNA and M147, F150, and Y151 form the three prongs of the hydrophobic plug that binds to the PCNA socket.

**Figure 2 ijms-23-11099-f002:**
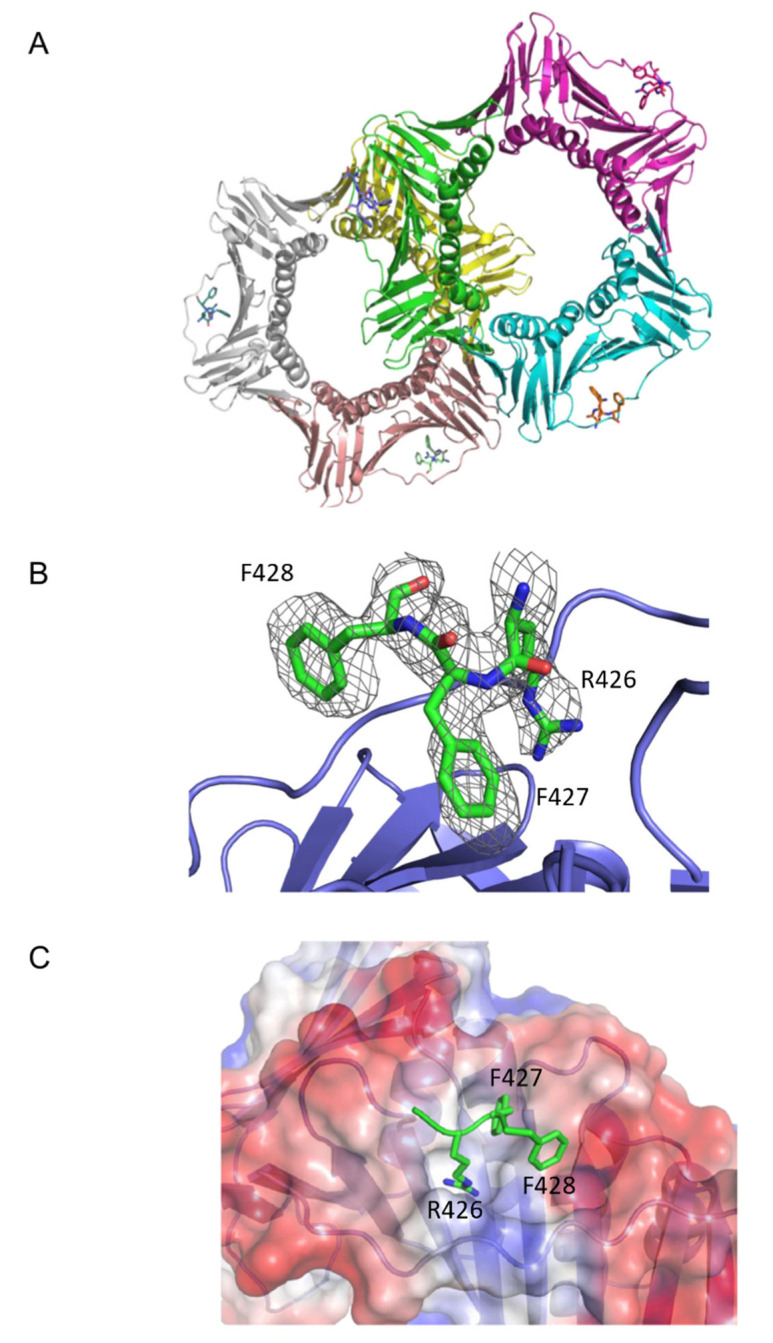
CAF-1 p150 PIP binding to PCNA. (**A**) The two PCNA homo-trimers are present in the asymmetric unit. Each PCNA monomer is bound to a PIP. Three CAF-1 p150 PIP residues (RFF) are depicted in stick representation. PIP binding occurs on a hydrophobic surface located between the interdomain connecting loop (IDCL) and the underlying β-sheet of PCNA. (**B**) p150 PIP (green) is shown in stick representation, with F0-dFc omit electron density map contoured at 1.0σ covering the p150 residues. Only the RFF residues are shown in the model. PCNA is shown in blue. (**C**) The p150 PIP interacts with a hydrophobic surface of PCNA. Three residues of the CAF-1 p150 PIP, Arg 426, Phe 427, and Phe 428 are shown in stick representation (green). PCNA is represented with surface electrostatics calculated to a solvent radius of 1.5 Å. Red corresponds to negatively charged residues, blue to positively charged residues, and white to an uncharged surface.

**Figure 3 ijms-23-11099-f003:**
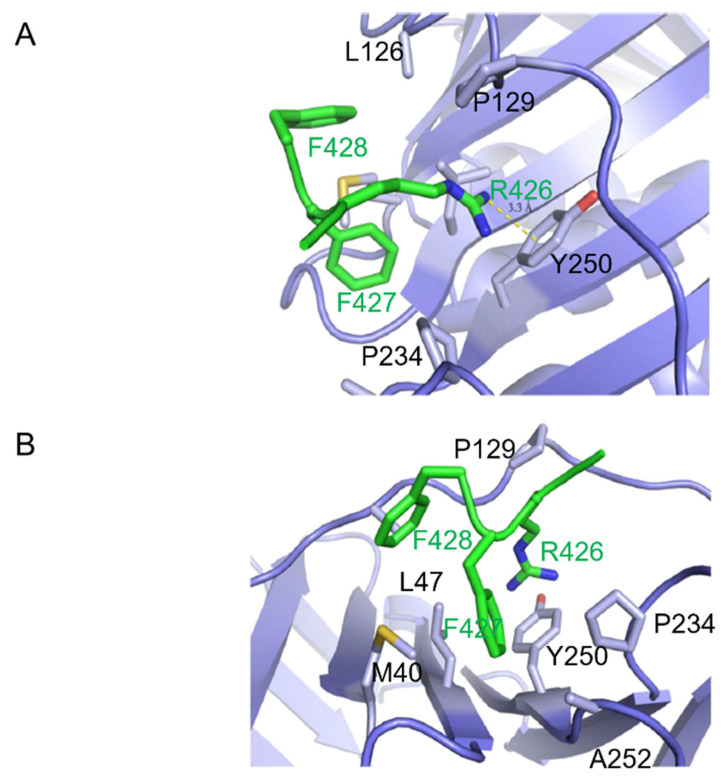
Cation-π interaction and orientation of PIP binding with respect to PCNA. (**A**) CAF-1 p150 PIP Arg 426 forms a cation-π interaction with Tyr 250 of PCNA. The side chain of PCNA Pro 234, Tyr 250, Ala 252, Met 40, Leu 126, Pro 129, and Leu 47 are shown in stick representation (light blue). Side chains of p150 PIP are shown in stick representation (green). Arg 426 of PIP extends into the hydrophobic pocket on the surface of PCNA and forms a cation-π interaction with Tyr 250. Bond length is measured between the terminal nitrogen of R426 and the center of the aromatic ring of Tyr 250. (**B**) Hydrophobic interaction between PCNA and p150 PIP. The side chains of the PCNA residues listed in (**A**) are shown in stick representation (light blue). The p150 PIP side chains are shown in stick representation (green). Phe 427 of the p150 PIP nestles into a hydrophobic surface of PCNA composed of Pro 129, Tyr 250, Leu 47, and Met 40 residues. Phe 428 of the p150 PIP is less deeply buried in the hydrophobic surface of PCNA. (**C**) Paths of polypeptide chains when the PIPs of different proteins are bound to PCNA. The drawing is derived from the co-crystal structures of PIP peptides bound to PCNA. Pol δ p66-PCNA (dark red) [20], Pol κ-PCNA (red) [20,28], p21-PCNA (magenta) [19], and CAF-1 p150 PIP2-PCNA (green) were aligned to FEN1-PCNA (salmon) [20], with root mean square deviation of <0.6 Å for all alignments. For clarity, PCNA is hidden. All the PIPs adopt a short 3_10_ helix conformation when bound to PCNA, except the CAF-1 p150 PIP, which assumes a different orientation with respect to PCNA. Cation-π interaction and orientation of PIP binding with respect to PCNA. (**D**) p150 PIP binding to PCNA is compared with the canonical PIPs of p21 and FEN1. The p21 PIP and p150-PIP were aligned to the FEN1 PIP with root mean square deviations of 0.55 and 0.50, respectively. The table shows key residues of the canonical PIPs and the corresponding residues of the CAF-1 p150 PIP in bold. The conserved glutamine replacement by lysine in the p150 PIP is indicated in blue. The underlined residues are those shown in the stick representation. FEN1 bound to PCNA (salmon); p21 bound to PCNA (magenta); and p150 PIP bound to PCNA (green). The glutamines in p21 and FEN1 are interact with the Q pocket on PCNA (visible in the top left-hand corner). Residues M146, F149, and Y150 in p21 and residues L341, F343, and F344 in FEN1 form 3_10_ helices and occupy the hydrophobic pocket on the surface of PCNA. In contrast, R426 and F427 of CAF-1 p150 both occupy the hydrophobic pocket. Strikingly, R426 (p150) overlaps with Y150 (p21) and F344 (FEN1), which is unusual because R is not a conserved residue in canonical PIPs. F427 and F428 of p150 are both conserved in canonical PIPs but shown to occupy hydrophobic regions not expected for canonical PIP residues. F427 of p150 overlaps with L341 (FEN1) and M146 (p21) but, given their conservation, not the expected F344 (FEN1) or Y150 (p21). F428 of the p150 RFF occupies a hydrophobic patch with no overlap with conserved residues in canonical PIPs but instead overlaps with a non-conserved residue, V348 (FEN1).

**Figure 4 ijms-23-11099-f004:**
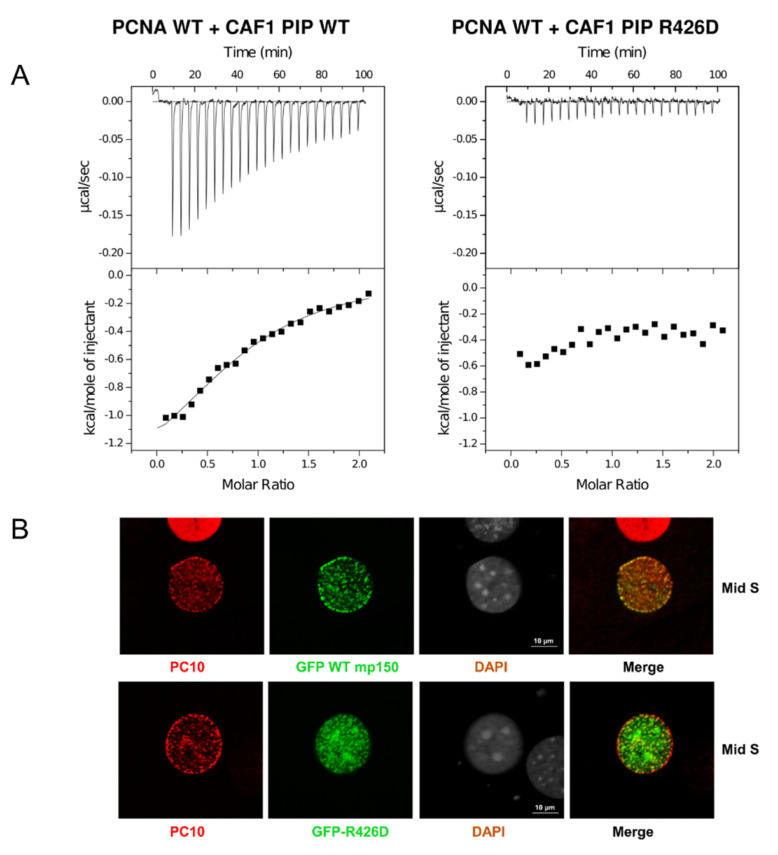
Cellular and molecular consequences of the PIP-Arg 426 mutation. (**A**) In ITC experiments, the CAF-1 PIP wild-type and R426D substituted peptides were progressively titrated into a sample cell containing PCNA. The upper half of each panel shows the measured heat exchanges following each peptide injection. The lower half of each panel shows the enthalpic changes as a function of the molar ratio of peptide to PCNA monomer. The black squares correspond to individual injections. Titrations were performed in 10 mM sodium phosphate, pH 7.0, 10 mM NaCl, and at 23 °C. (**B**) The Arg426 mutation causes mis-localization of mouse p150 (mp150) in mid S-phase cells. Mouse NIH3T3 cells were transduced with constructs for expression of GFP CAF1 p150 WT or R426D mutant and stained with the PC10 monoclonal antibody (PC10) against PCNA to detect DNA replication foci. The mis-localization of the R426D mutant is most evident in mid S-phase cells, where the red PCNA staining is, in part, localized to the periphery of the cell nucleus, whereas the green CAF-1 p150 R426D is more localized to the pericentric heterochromatin (large foci seen with DAPI staining). (**C**) Arg426 is required for nucleosome assembly during SV40 DNA replication. Left panel: CAF1 p150 wild-type and R426D mutant proteins were expressed in the rabbit reticulocyte lysate in the presence of [^35^S]-methionine and detected by SDS-PAGE and autoradiography. Middle panel: Total DNA from nucleosome assembly reactions was stained with ethidium bromide. rCAF-1 is purified full-length recombinant CAF-1 added as a positive control. Right panel: Nucleosome assembly reactions were performed in the presence of increasing amounts of wild-type p150 PIP or R426D mutant. Replicated DNA was detected by incorporation of [α-32P] dATP and autoradiography. Lane 1 corresponds to a negative control reaction lacking p150 where there was very little supercoiling observed.

**Figure 5 ijms-23-11099-f005:**
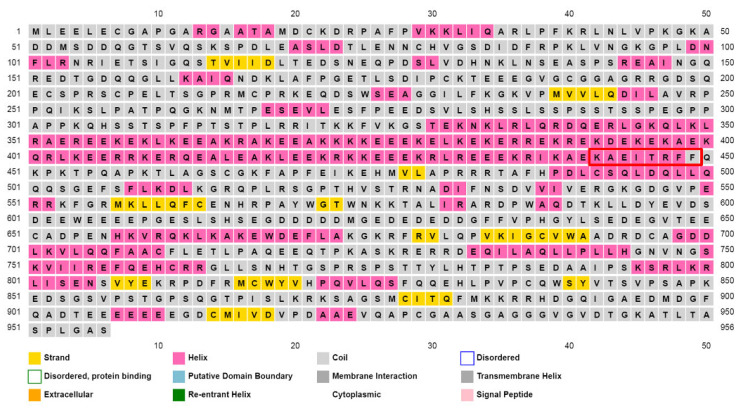
Secondary structure prediction analyses of human CAF-1 p150.

**Figure 6 ijms-23-11099-f006:**
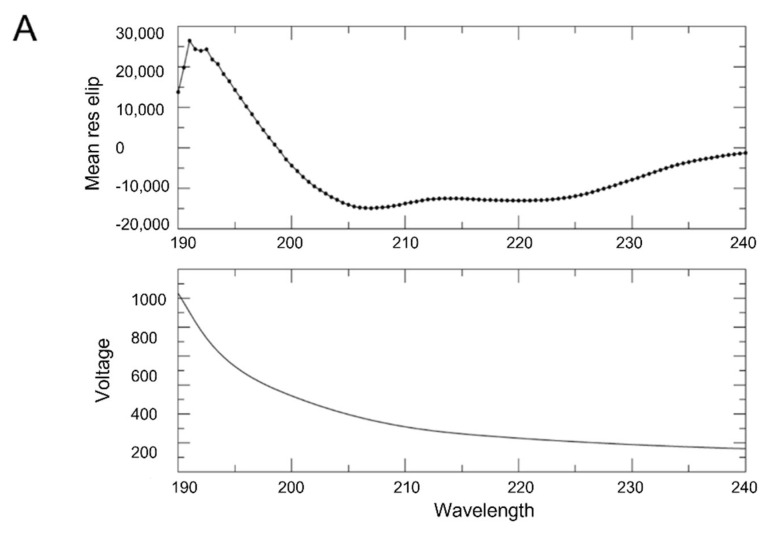
CD and SEC-MALS analyses of p150L. (**A**) The CD spectra indicate that CAF-1 p150L is composed mainly of α-helices. CD spectrum was acquired in 10 mM sodium phosphate, pH 7.2, 10 mM NaCl, and 0.1 mM TCEP buffer with a data pitch of 0.5 nm over a range from 260 nm to 190 nm. (**B**) SEC-MALS analysis. The p150L protein was injected onto a size exclusion chromatography column coupled to multi-angle light scattering to compute the molar mass and oligomerization status of the protein (blue trace). The principal peak of p150L corresponds to a monomer (19 kDa), whereas the minor peak may form a dimer (30–40 kDa). Relative to the larger 67 kDa BSA monomer (~13.7 mL, red trace), the early-eluting position of the minor peak is indicative of a non-globular shape. The dimeric configuration is most likely not a coiled-coil due to the drastically different elution volume than the monomeric p150L. (**C**) AlphaFold-Multimer predicts an elongated coiled-coil arrangement of p150L. The models are colored by confidence (pLDDT) scores, and the RFF residues are shown as sticks. The scale bar is 170 Å.

**Figure 7 ijms-23-11099-f007:**
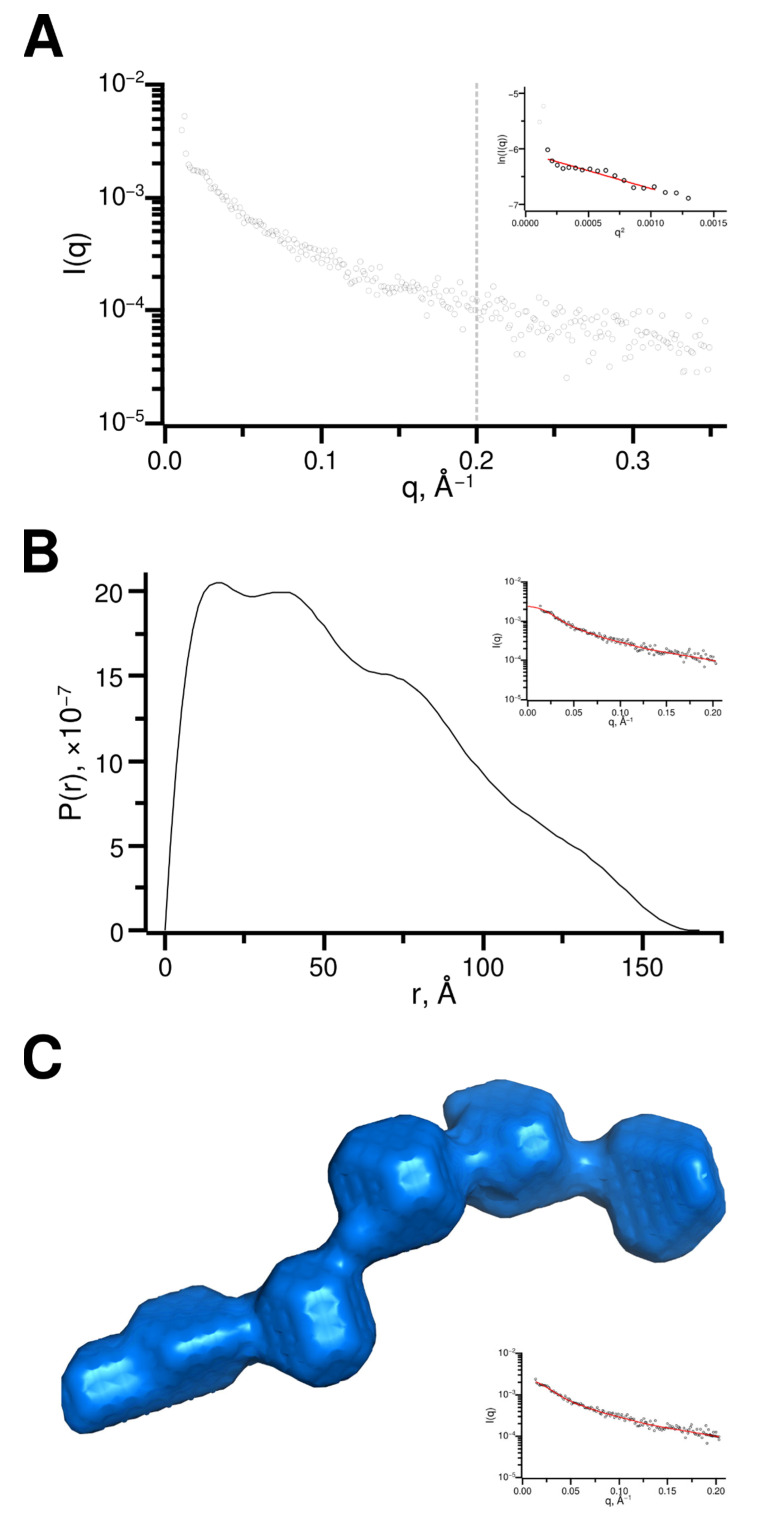
SEC-SAXS data analysis (**A**) Scattering curve representing the averaged scattering from frames 96 to 108, as highlighted in Appendix A. Data beyond q of 0.2 Å^−1^ (grey dotted line) was excluded from the rest of the analysis. The Guinier region is shown in the inset. (**B**) Pair distribution (P(r)) function determined using GNOM and manual adjustments to achieve a probability function that gradually approaching zero at maximum dimension. The fit to experimental data is shown in the inset. (**C**) Surface representation of an ab initio CAF-1 p150L model as calculated by DAMMIF and further refined by DAMMIN. The fit to the experimental data is shown in the inset. Scattering data has been deposited in the Small Angle Scattering Biological Data Bank (SASBDB) under the accession number SASDP79.

**Figure 8 ijms-23-11099-f008:**
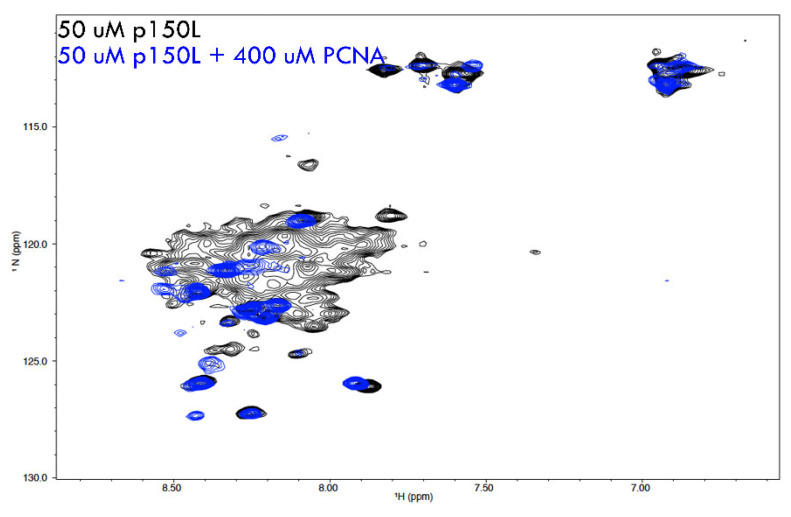
Binding of [15N]-p150L to PCNA studied by nuclear magnetic resonance.

**Figure 9 ijms-23-11099-f009:**
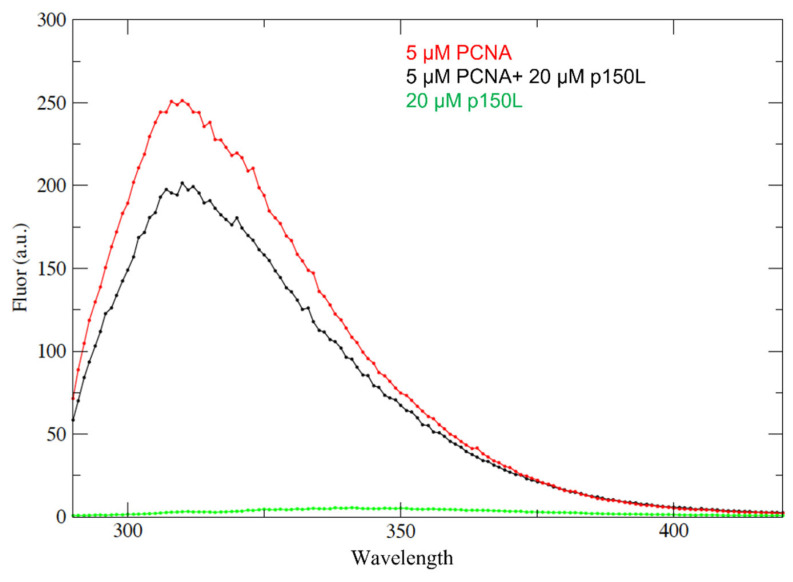
p150L binding to PCNA studied by fluorescence quenching. Samples were excited at 280 nm and the emission spectra recorded between 295 nm and 420 nm were collected. The red curve shows the emission spectrum for 5 µM PCNA monomer on its own. The black curve shows the spectrum for a complex containing 5 µM PCNA monomer and 20 µM p150L. The green curve shows the spectrum for 20 µM p150L on its own. Due to the absence of tryptophan, the emission spectrum for p150L is near the baseline.

**Table 1 ijms-23-11099-t001:** Summary of crystallographic statistics.

PCNA-p150(PIP2)	
**Data Collection**	
Space Group	C121
a, b, c (Å)	143.94, 83.24, 145.04
α, β, γ (°)	90.00, 107.32, 90.00
Resolution (Å)	46.2–2.4
Measured Reflections	248,112
Unique Reflections	64,159
Average Redundancy	3.9
Completeness (%)	100%
R_merge_ (Outer Shell)	0.073 (0.436)
Mean I/σ _(I)_ (Outer Shell)	7.9 (2.8)
**Refinement**	
R_work_	0.2318
R_free_	0.3044
No. Reflections (R_work_)	63,651
No. Reflections (R_free_)	3223
% R_free_	5.06

Values in brackets correspond to outer shell, resolution 2.53–2.40 Å; σ(I) is the standard deviations (SD) of the measured intensity (I); R_free_ is Σ| I-<I>I/Ʃ| where <I> is the mean intensity of all observations; R_work_ is Σ| F_o_-F_c_ I/ƩF_o_ for all data excluding data to calculate R_free_; R_free_ is Σ| F_o_-F_c_ I/ƩF_o_ for all data excluded from refinement.

**Table 2 ijms-23-11099-t002:** Thermodynamic parameters of PCNA binding to PIP ^a^.

Exptl Buffer and Peptide	Kd (μM)	N ^b^	ΔG (kcal mol^−1^)	ΔH (kcal mol^−1^)	ΔS (cal K^−1^ mol^−1^)
10 mM Na-phosphate (pH 7.0), 10 mM NaCl PCNA wild type + CAF-1 PIP wild type	24	0.83	−6.25	−1.868	14.8
PCNA wild type + CAF-1 PIP R426D	No binding				

^a^ Determined by isothermal titration calorimetry at 296 K. All data are derived from Figure 4C, and the binding curves were fit to one binding site per PCNA monomer, as observed in several crystal structures of PIPs bound to PCNA. ^b^ Stoichiometry of peptide binding per PCNA monomer.

## Data Availability

Scattering data has been deposited in the Small Angle Scattering Biological Data Bank (SASBDB) under the accession number SASDP79.

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
