# Peer review of "Unorthodox PCNA Binding by Chromatin Assembly Factor 1"

_ijms, 2022, doi:10.3390/ijms231911099_

Round 1

Reviewer 1 Report

In this manuscript, the authors have provided structural insight into the binding of the CAF-1 PIP domain with PCNA. I found the data convincing and the manuscript well-written. I have only the following two comments:

·       In your structural figures, please consider labelling important residues and features. For example, I would have found it quicker and easier to interpret Figure 1C if the Q-pocket of PCNA, and residues Q144, M147, F150 and Y151, were labelled.

·       In section 2.2.1., the authors create a PCNA-Y250I mutant, which they claim does not bind to the CAF-1 PIP. This is an important experiment to verify their model. I do not know why the authors have chosen not to include these data, but I do ask them to reconsider.

Author Response

  1. In your structural figures, please consider labeling important residues and features. For example, I would have found it quicker and easier to interpret Figure 1C if the Q-pocket of PCNA, and residues Q144, M147, F150, and Y151, were labeled.

Response1: I would like to thank reviewer 1 for their kind words. As per the suggestion from Reviewer 1, the residues were labeled in the corresponding figures in the manuscript.

  1. In section 2.2.1., the authors create a PCNA-Y250I mutant, which they claim does not bind to the CAF-1 PIP. This is an important experiment to verify their model. I do not know why the authors have chosen not to include these data, but I do ask them to reconsider.

Response2: As per the suggestions from Reviewer 1, I have included the ITC experiment with PCNA-Y250I mutant and CAF-1 PIP-WT as supplementary figure 1.

Reviewer 2 Report

The manuscript, "Unorthodox PCNA binding by Chromatin Assembly Factor 1" by Nair and colleagues explores the molecular and structural mechanisms that regulate the binding of PCNA to CAF-1 using a variety of techniques including crystallography and mutagenesis. In general this is a high quality manuscript, with a well written introduction, detailed and clear presentations of results and conclusions that are well supported by the data. I'm happy to recommend this manuscript for publication in IJMS.

Author Response

Response: I would like to thank reviewer 2 for their encouraging words.

Reviewer 3 Report

In this manuscript, Amogh Gopinathan Nair et al investigate the interaction of chromatin assembly factor CAF1 to PCNA, an essential factor for nucleosome formation behind the replication fork. Like another PCNA-interacting proteins, CAF1 p150 has a PIP-box, but it has an unusual Arg just before the two aromatic residues, FF. The crystal structural analysis revealed that its binding mode to the hydrophobic pocket on the surface of PCNA was different from those of usual PIP-peptides, mediated by a cation-pai interaction created between Arg of CAF1 p150 and Tyr of PCNA. Arg to Asp change caused a failure of CAF1 recruitment to replication foci. CAF1 p150 PIP box is located at the C-terminal end of long alpha helix domain. This domain has a DNA binding activity and appears to form an oligomeric coiled coil structure. Together the authors propose that helix domain and PIP-box of p150 work together to coordinate the DNA replication to nucleosome formation. The authors carried out substantial works in this paper. This reviewer requests some more information and careful analysis before acceptance. 

1. The RFF amino acid portion of CAF1 p150 PIP box associates with the hydrophobic pocket of PCNA, but its binding mode is quite different from those of usual PIP peptides. The authors suggest that this unique binding mode, though its affinity to PCNA is low, is important for CAF1 to function at the replication fork. This is an intriguing finding in this paper. However, (1) readers will wonder if this binding mode is unique to CAF1 p150 or more expanded? In a list of PCNA interaction partners (ex. Prestel et al, Cellular and Molecular Life Science, 2019), RFF-type proteins, such as Msh3, are noticed. On the other hand, yeast CAF1s do not have R at the corresponding position. Drosophila and worm CAF1 p150 have K instead of R, an amino acid with a similar property. In addition, (2) authors suggest that R residue is critical not only for recruiting to PCNA, but for coordinating PCNA-dependent multiple steps. To confirm such an idea, it is important to replace the CAF1 p150nPIP box with a canonical one and see its effect on replication.  

2. The R to D displaced CAF1 p150 PIP sequence is KAEITDFFQKPKT. As underlined, the motif fits with PIP-degron, a degron for CRL4Cdt2 ubiquitin ligase, targeted for ubiquitylation when bound on PCNA. TD motif confers a high affinity to PCNA (Havens et al, Molecular Cell, 2009). It is possible that the R to D changed CAF1 will be degraded at the replication sites, and thus not detected at the PCNA foci, but those recruited to heterochromatin were not degraded, resulted as shown in Fig. 4B. 

3. The purified DNA free CAF-1 p150L was analyzed by CD and SEC-MALS, and examined for PCNA binding. The authors found oligomeric, mostly trimeric, forms of p150L peptides. Unfortunately, DNA binding activity of neither monomeric nor multimeric forms of p150L was examined. It is hard to imagine how the multimeric forms of CAF1 coordinate DNA-binding and histone deposition. The oligomeric forms might be an artefact formed only when p150L portion was expressed, because CAF1 appears to be present as a single trimeric form (such as in Sauer et al, eLife, 2017).  

4. Figure 5. Secondary structure prediction analysis suggests a presence of a long helix structure in p150 and the p150 PIP box located at its C-terminal end. On the other hand, this region is predicted to take a disordered domain (Sauer et al. NAR, 2018). (1) Analysis with an Alphafold2 program will support to predict a reliable structure. (2) If p150 PIP box was located at the end of helix domain, the N-terminal region upstream from the p150 PIP box would take a fixed structure. However, no defined structure was detected in both sides extended from p150 PIP box in the crystal structure analysis (Figure 2). Authors need to explain such an anxiety. 

5. Figure 5. In the text in lane 120-121, RFF residues are numbered as 426-428. However, these residues are not detected at corresponding positions in Fig 5, instead they are found at residues 446-448. 

6. Readers are acknowledged if PIP boxes are marked in the Supplementary Figures1 to 3 as red boxes as shown in Figure 5.

Author Response

In this manuscript, Amogh Gopinathan Nair et al investigate the interaction of chromatin assembly factor CAF1 to PCNA, an essential factor for nucleosome formation behind the replication fork. Like another PCNA-interacting proteins, CAF1 p150 has a PIP-box, but it has an unusual Arg just before the two aromatic residues, FF. The crystal structural analysis revealed that its binding mode to the hydrophobic pocket on the surface of PCNA was different from those of usual PIP-peptides, mediated by a cation-pi interaction This reviewer requests some more information and careful analysis before acceptance. created between Arg of CAF1 p150 and Tyr of PCNA. Arg to Asp change caused a failure of CAF1 recruitment to replication foci. CAF1 p150 PIP box is located at the C-terminal end of long alpha helix domain. This domain has a DNA binding activity and appears to form an oligomeric coiled coil structure. Together the authors propose that helix domain and PIP-box of p150 work together to coordinate the DNA replication to nucleosome formation. The authors carried out substantial works in this paper.

  1. The RFF amino acid portion of CAF1 p150 PIP box associates with the hydrophobic pocket of PCNA, but its binding mode is quite different from those of usual PIP peptides. The authors suggest that this unique binding mode, though its affinity to PCNA is low, is important for CAF1 to function at the replication fork. This is an intriguing finding in this paper. However, (1) readers will wonder if this binding mode is unique to CAF1 p150 or more expanded? In a list of PCNA interaction partners (ex. Prestel et al, Cellular and Molecular Life Science, 2019), RFF-type proteins, such as Msh3, are noticed. On the other hand, yeast CAF1s do not have R at the corresponding position. Drosophila and worm CAF1 p150 have K instead of R, an amino acid with a similar property. In addition, (2) authors suggest that R residue is critical not only for recruiting to PCNA, but for coordinating PCNA-dependent multiple steps. To confirm such an idea, it is important to replace the CAF1 p150nPIP box with a canonical one and see its effect on replication.

I would like to thank reviewer 3 for their questions and I have done the best of my ability to provide the clarifications.

Response 1a) The reviewer has correctly pointed out that mutS homolog 3 or the MSH3 is a part of the DNA mismatch repair mechanism which has the three residues RFF in their PIP box. The entire PIP box sequence of Msh3 is 19-APARQAVLSRFFQSTGSLKSTSSST-41 and in accordance with the canonical PIP box sequence the first residue corresponds to a glutamine whereas in the case of CAF-1 PIP, the conserved PIP box glutamine is replaced by a lysine at this position, and this makes the binding mode of CAF-1 p150 non-canonical and unique in the case of human sequence 442-KAEITRFF-449 and mouse sequence 421-KAEITRFF-428. The reason why we think the non-canonical PIP box of CAF-1 is unique is due to the fact that it hasn’t caught our attention of a similar PIP box structure where the starting residue is a lysine followed by the presence of arginine and two phenyl alanine’s. Here I would like to show an example of the crystal structure of a canonical - p21 PIP (yellow) and non canonical error prone polymerase kappa (cyan) with different binding patterns.

In canonical p21 PIP, highly conserved glutamine crucial for PCNA binding is occupying the glutamine Q-pocket whereas this is replaced by a lysine in the noncanonical pol kappa, and this lysine is not making any interactions with PCNA and is pointing away into the solvent. This depicts that there are differences in the way a canonical vs non-canonical PIP box protein mediates its binding on to the PCNA surface.

Response1b) Another surprising thing about the CAF-1 sequences is that in multiple yeast species it is canonical with a conserved glutamine in the first position as mentioned in line 1034 of the submitted publication - for example in S. Cerevisiae it is QSRIGNFF. Despite the presence of a canonical PIP box, the secondary structure prediction also shows the presence of a conserved long alpha helix in the S. Cerevisiae and S. pombe. The PIP box sequences are also analysed in multiple candida species to be canonical as shown in the following figure. So, currently I can only say that the binding mechanism in the fungal species might be regulated differently, and further studies are necessary to understand why this is so.

Response 1c) In the case of Drosophila and worm CAF1 p150 in addition to having K instead of R, both amino acids with positive charge, the first residue of the PIP box is neither a glutamine nor lysine. This is something which I cannot explain why this is the case with CAF-1 p150 in these two species.

Response1d) To address the reviewer’s idea of making the CAF1 p150 PIP box a canonical one, a previous research article from our lab (PMID: 19822659) has shown that substitution of the CAF-1 p150 lysine by a glutamine as to mimic a canonical PIP-box sequence enhances the direct binding of CAF-1 p150 PIP to PCNA at moderate ionic strength (Fig-7 and Table 1). In addition, a synthetic peptide of p150 with lysine to glutamine substitution could deplete PCNA from S100 extracts. This signifies the importance of having a non-canonical lysine at the starting position of the CAF-1 PIP box to avoid its non-specific binding to PCNA and the presence of an arginine residue that mediates a unique cation-pi interaction with the tyrosine 250 of PCNA as we observed in the crystal model.

  1. The R to D displaced CAF1 p150 PIP sequence is KAEITDFFQKPKT. As underlined, the motif fits with PIP-degron, a degron for CRL4Cdt2 ubiquitin ligase, targeted for ubiquitylation when bound on PCNA. TD motif confers a high affinity to PCNA (Havens et al, Molecular Cell, 2009). It is possible that the R to D changed CAF1 will be degraded at the replication sites, and thus not detected at the PCNA foci, but those recruited to heterochromatin were not degraded, resulted as shown in Fig. 4B.

Response 2) The reviewer has correctly pointed out the fact that the R to D displaced CAF1 p150 PIP sequence is KAEITDFFQKPKT. The motif for PIP-degron is QXXᴪTDөөXXXB. The most striking five features of a PIP degron are a conserved glutamine is present at position 1(Q), an aliphatic residue at position 4 (leucine, isoleucine, valine or methionine), TD residues that gives higher PCNA binding specificity, two aromatic residues өө at positions 7 and 8 (phenylalanine, tyrosine or tryptophan) and a basic residue either lysine or arginine as the 4th residue after the two aromatic residues. This is represented in the following schematic for CRL4Cdt2 ubiquitin ligase. If this PIP degron motif is present in the PIP box, this will cause targeted degradation as the reviewer correctly states. But since in the case of CAF-1 p150 PIP, the first feature is not fulfilled and as the glutamine is replaced by lysine this might not recapitulate the formation of a PIP degron despite its R to D mutation and thus making it unlikely that this would be recognized for PIP-degron mediated degradation.

  1. The purified DNA free CAF-1 p150L was analyzed by CD and SEC-MALS and examined for PCNA binding. The authors found oligomeric, mostly trimeric, forms of p150L peptides. Unfortunately, DNA binding activity of neither monomeric nor multimeric forms of p150L was examined. It is hard to imagine how the multimeric forms of CAF1 coordinate DNA-binding and histone deposition. The oligomeric forms might be an artefact formed only when p150L portion was expressed, because CAF1 appears to be present as a single trimeric form (such as in Sauer et al, eLife, 2017).

Response 3) The purified CAF-1 p150L was analyzed in a CD experiment to show that the expressed protein is helical as per the secondary structure prediction. Based on several lines of evidence and as mentioned in Sauer et al, eLife, 2017, we acknowledge that new (H3-H4)2 tetramers are assembled from two CAF-1 trimeric complexes that each bind to an H3-H4 dimer. We thank the reviewer 3 for this insightful observation into p150L binding DNA. Previous studies from  (Sauer et al, eLife, 2017) has also demonstrated that the KER domain is reported to have DNA binding activity (40bp) (Sauer et al, eLife, 2017). Indeed, the highly charged KER domain can bind DNA and we subsequently found out that some DNA remained bound to our purification despite 2.5 M NaCl washes during cell lysis and affinity chromatography. More recently we added a benzonase digestion and further 1 M NaCl washes to remove the DNA contamination. The clean p150L sample was analyzed again by SEC-MALS and SEC-SAXS, resulting predominantly in a monomer. We have replaced Figure 6B in the manuscript and modified the text to reflect these new data. 

Several studies also indicate that the purified shorter versions of CAF-1 p150 and cac1 have a tendency to oligomerize (dimerize) in the absence of their binding partners (PMID: 11296234 PMID: 16826239, PMID: 27690308, PMID: 22941638). As the reviewer correctly points out if we were to use the full-length CAF-1 p150 or the whole trimeric CAF-1 complex comprising  p150 (CHAF1A), p60 (CHAF1B), and RbAp48 (RBBP4) the results in solution might be different, but since this was not attempted in our studies I cannot comment on this.

  1. Figure 5. Secondary structure prediction analysis suggests a presence of a long helix structure in p150 and the p150 PIP box located at its C-terminal end. On the other hand, this region is predicted to take a disordered domain (Sauer et al. NAR, 2018). (1) Analysis with an Alphafold2 program will support to predict a reliable structure. (2) If p150 PIP box was located at the end of helix domain, the N-terminal region upstream from the p150 PIP box would take a fixed structure. However, no defined structure was detected in both sides extended from p150 PIP box in the crystal structure analysis (Figure 2). Authors need to explain such an anxiety.

Response 4.) The reviewer is correct that there exists a discrepancy between the DISOPRED (used in Sauer et al. NAR 2018) and PSIPRED servers. As the reviewer correctly points out from previous studies and Sauer et al. NAR, 2018, the CAF-1 p150 has multiple disordered regions as seen from the following figures (http://bioinf.cs.ucl.ac.uk/psipred/&uuid=f1ca9cd4-1a73-11ed-9451-00163e100d53). Inspite of the presence of disordered regions as we observed in our analysis the predicted helix region is terminating right after the CAF-1 p150 PIP box residues.

Response 4.1) In addition, as per the suggestion from reviewer 3, we performed an AlphaFold2 model prediction (AlphaFold Protein Structure Database (ebi.ac.uk) see entry Q13111). Indeed, AlphaFold2 predicts a long alpha helix covering residues 335-448, matching almost exactly the predicted helix for residues 330-448. We have added this support to the manuscript (lines 357). Furthermore, we used AlphaFold-Multimer to evaluate the possibility that the KER domain does indeed form a coiled-coil, as it was predicted for some time (Figure 6C line 391). The arginine residue is highlighted in the following picture.

Response 4.2)  The lack of an apparent helix N-terminal of the co-crystallized RFF residues may probably be due to not having enough amino acids to form a helix. Both secondary structure prediction, in silico modeling and in solution SAXS data demonstrate that the p150L domain of CAF1 adopts a long alpha-helical structure. However, it cannot be excluded that upon binding to PCNA, part of this domain, including the PIP box, rearranges in order to exhibit its residues involved in the binding interface. In other words, it is possible that binding of CAF-1 p150L to PCNA would stabilize the N-terminus of this structure as a sort of higher order regulation which might involve multiple post translational modifications on either of these proteins. While understanding this mechanism is important, it is beyond the scope of this manuscript.

  1. Figure 5. In the text in lane 120-121, RFF residues are numbered as 426-428. However, these residues are not detected at corresponding positions in Fig 5, instead they are found at residues 446-448.

Response 5) The CAF-1 p150 sequences in human and mouse are identical near the PIP box, but the total length of Q13111 in CAF1A_HUMAN is 956aa and total length of Q9QWF0, CAF1A_MOUSE is 911 aa. Because there are extra residues in the full length human CAF-1 sequences, the amino acid numberings differ by an extra 21 amino acids just before the start of the PIP box. The crystallography studies and the biochemistry experiments were performed considering the mouse sequences and hence in the mouse sequence the PIP box begins at 421-KAEITRFF-428, this was mentioned in section 2.2.1 line number 212 and section 4.1 line 638. The secondary structure prediction, represented in Figure 5 and subsequent SEC-MALS, SEC-SAXS experiments were performed with human CHAF1A sequences and correspondingly PIP box sequences in the case of human will be 442-KAEITRFF-449 this was the reason for these differences, this was briefly mentioned in line 344 and in supplementary method section S.M.2 line 866.

  1. Readers are acknowledged if PIP boxes are marked in the Supplementary Figures1 to 3 as red boxes as shown in Figure 5.

Response 6) The suggested changes from reviewer 3 were incorporated in the supplementary figures of the manuscript.